# Supply Chain Power and Corporate Environmental Responsibility: Mediation Effects Based on Business Performance

**DOI:** 10.3390/ijerph18179264

**Published:** 2021-09-02

**Authors:** Tao Luo, Ruhe Xie

**Affiliations:** 1School of Economics and Statistics, Guangzhou University, Guangzhou 510006, China; 1111764005@e.gzhu.edu.cn; 2School of Economics and Management, Guangdong Construction Polytechnic, Guangzhou 510440, China; 3School of Management, Guangzhou University, Guangzhou 510006, China

**Keywords:** supply chain power, environmental responsibility, business performance, mediation effect model

## Abstract

Corporate environmental responsibility is an important component of corporate social responsibility. Based on data from Chinese listed companies from 2010–2018, the relationship between the two was explored using a fixed-effects model, with the supply chain net cash ratio being used as a proxy variable for supply chain power. This study found that supply chain power has a significant positive impact on the environmental responsibility of upstream suppliers and downstream customers. There is a vertical spillover effect of supply chain power on corporate environmental responsibility. A stepwise regression approach was used to investigate the mediating effect of business performance between the two, and the bootstrap method was used to test the existence of the mediating effect. Firms can use their power in the supply chain to reasonably allocate profits in the supply chain, which not only helps them to fulfil their environmental responsibility but also influences other firms in the supply chain to fulfil their environmental responsibility.

## 1. Introduction

Business activities not only bring employment and drive economic development but also create huge environmental problems. More than 88% of consumers believe that companies should strive to achieve their business goals while improving society and the environment [1]. Discovering how companies can better fulfil their environmental responsibilities requires a clear understanding of the factors that influence their environmental responsibility. To this end, scholars have used stakeholder theory, high-level theory, environmental ethics theory, organisational reputation theory, and institutional theory to study the impact of various factors on corporate environmental responsibility at both the macro- and micro-levels, and to make corresponding recommendations from different perspectives.

However, the behaviour of firms is influenced not only by macro-factors or individual firm characteristics but also by the relationships among firms in the supply chain network. It is widely known that competition between enterprises has evolved into competition between supply chains. The behaviour of enterprises is no longer the behaviour of independent individuals, but the behaviour of mutual influence and interaction in the supply chain network. A company’s behaviour not only affects its own environmental responsibility but also the environmental responsibility of other enterprises in the supply chain. Environmental pollution incidents associated with well-known companies have had a serious negative impact on the finances and reputation of those companies and their supply chain partners and, more importantly, have caused significant environmental and ecological damage. For example, in 2015, Volkswagen’s emissions scandal caused it to lose a third of its market value and it also faced “billions of dollars in fines and other financial penalties”, which had a large impact on the stock prices of its supply chain partners [2]. Jacobs et al. (2020) stated that companies in responsible supply chains should be aware of their supply chain partners’ ethical, social, and environmental performance, and these factors should be important considerations when assessing supply chain partners. The supply chain partners can take action to prevent and manage irresponsible and unethical behaviour by large and influential players [2].

Although firms in a supply chain are interdependent, the relationships between them are not symmetrical. Powerful firms in the supply chain have a great deal of influence over other firms [3]. Supply chain power refers to the ability of firms in the supply chain to influence the decisions of other firms [4,5]. Most existing studies have discussed the impact of firms on environmental responsibility by considering firms as standalone individuals and do not take the impact of supply chain power into account. If the impact on environmental responsibility is only studied from the perspective of individual firms, the impact of inter-firm relationships on the environmental responsibility of firms in supply chain networks cannot be explained. In order to fill the gap in these studies, this research focuses on the following three issues: (1) the impact of supply chain power on corporate environmental responsibility, (2) whether there is a vertical spillover effect of supply chain power on corporate environmental responsibility, and (3) whether firm performance has a mediating effect between supply chain power and environmental responsibility.

The rest of this paper is arranged as follows: Section 2 contains the literature review, Section 3 presents the theoretical analysis and the research hypothesis, Section 4 outlines the model and data, Section 5 presents the empirical research results and analysis, Section 6 outlines the test for the mediating effect of business performance, and Section 7 contains the conclusion.

## 2. Literature Review

Given the huge impact of corporate activities on the environment, environmental responsibility has become an important issue that needs to be addressed in the process of sustainable corporate development. Liu Quan et al. (2019) argued that with rapid economic development, the related social issues are more prominent than ever, especially the issue of how to effectively encourage companies to take on more corporate social responsibility and maintain sustainable social development. This has become an urgent task for managers and researchers [6]. At the macro-level, corporate environmental responsibility is influenced by factors such as government regulations, institutions, laws, media, and marketisation. At the micro-level, the factors that influence corporate environmental responsibility are more diverse. It is generally accepted that corporate governance characteristics include size, ownership, asset concentration, and Chair–CEO duality. Factors such as management characteristics also have an impact on corporate environmental responsibility. At the market level, competition is an important factor affecting corporate eco-responsibility. Meng, Zeng, Xie et al. (2016) used an empirical test of 792 listed manufacturing firms in China from 2006–2008 and found that too much or too little competition at the firm level and too much or too little market power can lead to a reduction in corporate environmental responsibility, while corporate environmental responsibility is related to the competitive situation faced by the firm [7].

In addition to the influence of the macro- and micro-factors on corporate environmental responsibility, scholars have also argued that the interrelationships between firms affect environmental performance. Gu (2021) argued that firm sales activities have a positive effect on environmental performance, with innovation having a mediating effect between the two. This is known as the environmental performance spatial spillover effect. The corporate environmental performance of a firm is influenced by its neighbours [8]. However, these studies focused on the horizontal spillover effect of corporate environmental responsibility. Therefore, the vertical spillover effect between upstream and downstream firms in the supply chain needs further study.

As firms are in a supply chain network, they interact with each other. This complex relationship affects not only the profitability of firms but also the fulfilment of their environmental responsibility. Jacobs and Singhal (2020) stated that this work needs to be extended from the focal firm to partners and competitors up and down the supply chain by considering the impact of poor corporate social responsibility behaviour [1]. This is because firms in the supply chain can influence other firms in the supply chain through power. Amaeshi, Osuji, and Nnodim (2008) stated that the more powerful firms in the supply chain can exert some ethical pressure on the weaker party through codes of conduct, corporate culture, anti-pressure group campaigns, personnel training, and value repositioning [9].

Previous studies have discussed corporate social responsibility from the perspective of the supply chain. Liu, Quan, and Li et al. (2018) argued that the competition among enterprises has shifted from brand competition among enterprises, as in the past, to competition among the supply chains [10]. Therefore, considering corporate social responsibility in the supply chain is conducive to improving the competitiveness of enterprises. In the two-level planning model proposed by Hsueh (2015), the core firm in the supply chain is the one that determines the level of social responsibility and optimal performance through the cooperation of other firms in the supply chain. Therefore, there is a need to rationally allocate corporate social responsibility in the supply chain to prevent the “tragedy of the commons” [11]. Ni, Li, and Tang (2010) stated that in order to cope with the pressure of social responsibility, companies should introduce codes of conduct to ensure that companies in their supply chains are socially responsible [12]. In addition, incentives for corporate social responsibility investments can be provided through the supply chain structure. Lehar, Song, Yuan et al. (2020) stated that the supply chain structure that best incentivises corporate social responsibility investments depends on the interactions among vertical coordination of corporate social responsibility, free-riding, and offsetting capabilities [13].

Further research has focused on the impact between supply chain networks and environmental responsibility, given that environmental responsibility differs from other social responsibilities and has a wider scope of impact and more stakeholders. Chen and Chen (2017) argued that in the context of global warming, companies and their supply chain(s) are meeting the challenge of reducing greenhouse gas emissions [14]. Tsen, Islam, Karia et al. (2019) stated that many environmental issues occur within the supply chain networks of firms. We need to integrate environmental issues into supply chain operations [15]. The successful integration of economic, environmental, and social sustainability goals provides innovative ideas for supply chain and operations management. Aphonia, Sarkis, and Davarzani (2015) stated that sophisticated supply chain and operations management largely addresses the wide range of environmental and social issues faced by organisations today. Integrating ecological issues with supply chain activities to promote corporate environmental responsibility is an effective way to address environmental issues [16]. Wu et al. (2017) found that when flexible volume contracts and wholesale price incentive contracts are used, companies can effectively meet their social responsibilities without adversely affecting profits [17].

Supply chain power has an impact on the behaviour of upstream and downstream firms. Although corporate social or environmental responsibility has been studied from a supply chain perspective, two issues still need further exploration. The direction and extent of the influence of supply chain power on corporate environmental responsibility is unclear. The mechanism of the interactions between firms with different degrees of power in the supply chain network is unclear.

The possible innovative contributions of this paper include the following. Firstly, the impact on corporate environmental responsibility is studied from the perspective of supply chain power, enriching the relevant research on the factors influencing corporate environmental responsibility. Secondly, the mediating effect of firm performance on supply chain power and environmental responsibility is verified. Thirdly, a novel method of empirical research is introduced into supply chain research, extending the existing methods used in supply chain research.

## 3. Theoretical Analysis and Research Hypotheses

### 3.1. Supply Chain Power and Firms’ Environmental Responsibility

Supply chain power has a significant impact on the business activities of firms. Burt (2009) stated that a firm’s position within a network is essentially the firm’s power on the network, which plays an important role in the firm’s behaviour [18]. A firm with strong power in the supply chain has the advantage of resources, information, and control, which provide a better competitive advantage for the firm and help the firm to fulfil its environmental responsibility. Therefore, we propose the following hypothesis.

**Hypothesis** **1.***Supply chain power has a positive influence on the company’s environmental responsibility*.

### 3.2. Supply Chain Power and Supply Chain Environmental Responsibility

Supply chain power reflects the relationship between a particular firm and other firms in the supply chain and has a significant impact on the firm’s business behaviour. Jacobs et al. (2020) argued that the financial impact of a major corporate event can be transmitted through the supply chain. The main reason for this is the contagion and competition effect of the focal firm’s event; the magnitude of the contagion effect is related to the characteristics of the focal event [2]. Amaeshi, Osuji et al. (2007) stated that power among firms in the supply chain is a key factor affecting the allocation of responsibility in supply chain relationships [9]. Therefore, we propose the following hypothesis:

**Hypothesis** **2.**
*Supply chain power has a positive influence on the environmental responsibility of upstream and downstream firms in the supply chain, and there is a vertical spillover effect.*


### 3.3. The Mediating Effect of Business Performance

Supply chain power affects a firm’s business performance. Patatoukas (2011) argued that the causal relationship between customer base concentration and a supplier firm’s performance is consistent [19]. Irvine et al. (2015) used a recently extended supplier–customer relationship dataset to introduce the dynamic relationship lifecycle hypothesis and found that the relationship between customer base concentration and profitability was significantly negative in the first few years of the relationship. However, as the relationship matured, the relationship became positive [20]. Korcan & Patatoukas (2016) found that changes in customer concentration in the market affected the fundamentals of the supplier and the value of the firm’s stock. A firm’s business performance is a major factor influencing corporate environmental responsibility [21]. Frooman (1997) and Mengue et al. (2010) showed a positive relationship between financial performance and corporate social responsibility. A firm’s supply chain power influences its business performance through its advantages in terms of resources, information, and control, and higher business performance contributes to better fulfilment of a firm’s environmental responsibilities [22,23]. Therefore, we propose the following hypothesis:

**Hypothesis** **3.**
*Business performance has a mediating effect between supply chain power and environmental responsibility.*


## 4. Model and Data

### 4.1. Sample

The dependent variable (*CER*) represents corporate environmental responsibility. The higher the *CER* score, the better the performance of the enterprise in fulfilling its environmental responsibility. The meaning of the indicator and the method for calculating it are shown in Table 1. The *CER* data were obtained from Hexun.com’s Social Responsibility Evaluation System.

The independent variable is supply chain power. Supply chain power refers to a company’s influence over the supply chain [24]. Supply chain power is represented by the net cash ratio (*CR*) as a proxy variable [25,26]. A company with supply chain power has the advantage of bargaining power in the process of trading. It can use its supply chain power to demand more operating capital from suppliers by increasing payables, reducing prepayments, extending payment schedules, or adhering to strict payment policies. For example, Dell used its supply chain power to extend payment terms for parts purchases, taking possession of large amounts of cash. The net cash ratio reflects the net cash assets of the enterprise’s operations. Therefore, it is reasonable to choose the net cash ratio to measure supply chain power. The stronger the net cash ratio, the stronger the firm’s position in the supply chain and the greater its supply chain power. The meaning and method of calculation of the indicator are shown in Table 1. Listed company data for the calculation of the *CR* indicator were taken from the China Securities Market and Accounting Research (*CSMAR*) database.

Based on existing studies, the following variables were selected as the main control variables: *Size* represents the size of the company; *Lev* represents the gearing ratio, which reflects the financial risk of the company; *Growth* represents the growth rate of operating income; *Dual* indicates whether Chair–CEO duality are the same person; *Top1* represents the percentage of shares held by the largest shareholder; *BM* represents the ratio of accounts to market capitalisation; *SOE* represents whether the company is a state-owned enterprise, reflecting the nature of ownership; *ListAge* represents the time of listing. The variables are calculated or defined in Table 1. The control variable data were obtained from the CSMAR database.

### 4.2. Model

The model of Hypothesis 1 was used to test the impact of supply chain power on corporate environmental responsibility. The explanatory variable is corporate environmental responsibility, and the explanatory variable is supply chain power (Model 1):(1)$$CER=C+\alpha_{1}CR+\beta_{i}Control+\varepsilon$$
where *i* represents an individual enterprise, *t* represents the year and Control stands for the controlled variables.

In order to study the vertical spillover effects in the supply chain, we constructed two matched samples, namely supply chain–producer and producer–customer samples. The CSMAR database provides data on the supply chain network relationship index. These data include supplier and customer data for listed companies. Firstly, the producers and suppliers were screened. The supplier and producer codes were obtained and then the relevant variables’ values were obtained from the CSMAR database. The environmental responsibility scores of the upstream suppliers were used as the dependent variable. Their related variables were used as control variables. The supply chain power of the downstream producer was used as the independent variable. Model 2 was then built, as shown below. Secondly, the manufacturers and customers were filtered. The manufacturer and customer codes were obtained, and the values of the relevant variables were obtained from the CSMAR database. The upstream producer’s environmental responsibility score was used as the dependent variable. The related variables were used as control variables and the supply chain power of the downstream customer was used as the independent variable (Model 3).
(2)$$CER^{supply}=C+\alpha_{1}CR^{producer}+\beta_{i}Control^{supply}+\varepsilon$$
(3)$$CER^{producer}=C+\alpha_{1}CR^{customer}+\beta_{i}Control^{producer}+\varepsilon$$
where *i* represents an individual enterprise, *t* represents the year and Control represents the controlled variables of the upstream firm.

In order to confirm Hypothesis 3, this study also used the mediating effect model to test the mediating effect of business performance between supply chain power and the company’s environmental responsibility (Model 4 and Model 5):(4)$$Performace=C+\lambda CR+\beta_{i}Control+\varepsilon$$
(5)$$CER=C+\alpha_{2}CR+\gamma Performance+\beta_{i}Control+\varepsilon$$
where *i* represents an individual enterprise; *t* represents the year; *Performance* represents the business performance of a company, which is reflected as the rate of return on total assets and the return on net assets; *ROA* is the rate of return on aggregate assets, which is obtained by dividing the net profit of the enterprise by the average balance of total assets; and *ROE* is the rate of return on net assets, which is obtained by dividing the net profit of the enterprise by the average balance of shareholders’ equity. The meanings of the other variables are the same as those in Model 1.

## 5. Regression Results and Analysis

### 5.1. Descriptive Statistics and Analysis

Table 2 presents the results of the descriptive statistics for each variable. In the study sample of this research, the mean corporate environmental responsibility score was 5.063, with minimum and maximum values of −15 and 30, respectively, and a standard deviation of 5.067. The mean value of the net cash ratio is 0.321, with minimum and maximum values of −1107.647 and 623.956, respectively, and a standard deviation of 12.439.

In order to verify the possible problem of multicollinearity in the model, correlation analysis was conducted on the variables in this study. The results showed that the correlation coefficients for most variables were less than 0.5. The correlation coefficients are summarized in Appendix A. The maximum test VIF value was 2.25. This indicates that there is no serious problem of multicollinearity in the regression model presented here.

### 5.2. Empirical Test Results

Due to the presence of outliers in the listed company data, all continuous variables were bilaterally tailed at the 1% quartile to eliminate the effect of extreme values. Hausman’s test was used to determine whether to apply a fixed effects model or a random effects model. The modified Hausman test yielded a *p*-value of less than 0.000 (χ^2^ (9) = 483.57), indicating that the empirical analysis required the use of a fixed effects model.

Table 3 shows the regression results for Model 1, testing the impact of supply chain power on corporate environmental responsibility. A valid sample of 17,356 was obtained after removing missing values. The influence coefficients of supply chain power on corporate environmental responsibility are 0.009 and 0.055, which have a positive effect and pass the 1% significance test. The results show that supply chain power has a significant positive effect on firms’ environmental responsibility scores. The reason for this may be that supply chain power reflects the overall competitiveness of a company. A company’s power in its supply chain network has a direct impact on its business performance. Strong supply chain power can lead to better business performance for a company, which then helps the company to better fulfil its environmental responsibilities.

Table 4 shows the regression results for Model 2, testing for vertical spillover effects between upstream suppliers and downstream producers. After matching via the method described above, 793 valid samples were obtained. The regression results, after controlling for time and industry effects, show that supply chain power has a positive effect on the suppliers’ environmental responsibility at the 1% significance level, validating the interactions between firms in the supply chain network, in which the producer can influence the environmental responsibility of its upstream suppliers.

Table 5 shows the regression results for Model 3, testing for vertical spillover effects between upstream producers and downstream customers. After matching via the method described above, 1408 valid samples were obtained. The results show that supply chain power has a positive impact on the environmental responsibility of upstream manufacturers at the 1% level of significance. This also validates the interactions between firms in the supply chain network, with downstream consumers having a positive impact on the environmental responsibility of the upstream producer.

These results show that the relationships between firms in a supply chain network have a corresponding impact. Stronger firms in the network can exert influence on less well-positioned firms in favour of corporate environmental responsibility in the supply chain, which validates the above hypothesis.

## 6. Test of the Mediating Effect of Business Performance

The arguments above showed that supply chain power has a significant impact on the environmental responsibility of companies. There are two main reasons for this. Firstly, supply chain power reflects a company’s competitiveness. The more concentrated a firm’s power in the supply chain, the more competitive the firm will be overall. Secondly, a firm’s strength in the supply chain indicates that the firm is likely to have more resources, information, and control, which are conducive to improving the firm’s business performance.

Model 4 was used to test the mediating effect of firm performance. In the model, γ was significant and represented whether the mediating effect of business performance is significant, and λ represents the direct effect of supply chain power on corporate environmental responsibility. In Model 4 and Model 5, λ* γ represent the indirect effect of a firm’s supply chain power on the firm’s environmental responsibility through the firm’s operating performance. If the mediating effect of business performance is present, the sign of λ should be significantly positive and the sign of γ should also be significantly positive.

The following table reports the regression results for Model 4. According to Table 6, supply chain power has a positive impact on the company’s return on total assets and return on net assets, with coefficients of 0.002 and 0.003, respectively, at the 1% level of significance. This is generally consistent with the findings of previous studies.

Table 7 shows the regression results for Model 5. Supply chain power had a positive effect on corporate environmental responsibility when the total return on assets was the mediating variable, with a coefficient of 0.006. When the return on net assets was the mediating variable, supply chain power had a positive effect on corporate environmental responsibility, with a coefficient of 0.008. However, these results did not pass the significance test. Whether there is a budding effect of corporate performance on supply chain power and corporate environmental responsibility needs to be tested further.

Table 7 shows that the stepwise regression methods did not identify any mediating effects. We used the Sobel method to test for mediation effects. When using the return on total assets and the return on net assets as mediating variables, the *p*-value was 0.0000, indicating the presence of a mediating effect at the 1% level of significance.

Bootstrapping has strong statistical power compared with other tests of intermediate effects and is recognised as an alternative to the Sobel method. It allows direct testing of the product of the coefficients. The bootstrapping method allowed the statistical significance of the mediating effect of firm performance to be tested. When the return on total assets was used as a mediating variable, the confidence interval was between 0.0194823 and 0.0377178. When the return on net assets was used as a mediating variable, the confidence interval was between 0.0136039 and 0.0392229. When both variables were used as mediating variables, their confidence intervals did not contain 0, which indicates that business performance has a mediating effect between supply chain power and environmental responsibility.

## 7. Conclusions

In the past, research on corporate environmental responsibility has been conducted from within to outside the firm, arguing that the implementation of corporate environmental responsibility activities affects the firm’s products and services, thus improving business performance. On the other hand, studies have argued that the pressures of factors external to the firm, such as laws, policies, market competition, consumers, etc., drive firms to implement environmentally responsible activities. In addition to these factors, the interactions between firms are also very important external factors that affect corporate environmental responsibility. The horizontal spillover effects between neighbouring firms have been studied. However, vertical spillovers from upstream and downstream enterprises have not received much attention. For example, companies may be required to have ISO 14000 environmental certification in order to become the suppliers of another firm. Therefore, the interaction between the upstream and downstream companies in the supply chain is also an important factor in the implementation of environmentally responsible activities by companies. This study constructed a panel data model for 2010–2018 using a sample of A-share listed companies in Shanghai and Shenzhen. The net cash ratio of the company was used as a proxy variable for supply chain power to discuss its impact on corporate environmental responsibility.

### 7.1. Theoretical Implications

Scholars have paid attention to the impact of external factors on corporate environmental responsibility. In fact, companies are no longer seen as independent individuals but are considered to interact with each other along the supply chain. Therefore, from a supply chain perspective, a firm’s implementation of environmental responsibility activities must be influenced by other firms in the supply chain. This study analysed the impact of supply chain power on environmental responsibility. The main findings are as follows.

Firstly, supply chain power has a clear positive impact on corporate environmental responsibility. In a supply chain network, the higher a company’s position in the supply chain network, the stronger its supply chain power and the more beneficial it is for companies to fulfil their environmental responsibility. Therefore, by occupying the core position in the supply chain network, enterprises can improve their business performance and help them fulfil their environmental responsibilities and achieve sustainable business operations.

Secondly, a firm’s supply chain power can have a beneficial impact on the implementation of environmentally responsible activities by other companies in the supply chain, i.e., there is a vertical spillover effect. This finding suggests that supply chain power enhances the peer effect of supply chain firms.

Third, it was verified that business performance has a mediating effect between supply chain power and corporate environmental responsibility. The reason for the vertical spillover effect of supply chain power on corporate environmental responsibility is that supply chain power affects the business performance of enterprises. In order to improve their business performance, enterprises are bound to implement environmental responsibility activities in accordance with the requirements of the enterprises with supply chain power. In a supply chain, the firm with supply chain power is generally the core firm in the supply chain.

In conclusion, this study found that the supply chain power of a company is beneficial to the implementation of corporate environmental responsibility activities. Moreover, a firm’s supply chain power has a vertical spillover effect on the implementation of environmental responsibility activities by other firms in the supply chain. Business performance plays an important mediating role between the two. These findings expand the field and scope of corporate environmental responsibility research. They enrich the theory of the external control of organisations from a supply chain perspective.

### 7.2. Managerial and Policy Implications

The findings have important managerial and policy implications. In terms of management, the core companies in a supply chain are generally the ones with supply chain power. The actions of core enterprises not only affect themselves but can also spill over vertically and affect other enterprises in the supply chain. If the core enterprises fail to effectively implement environmentally responsible activities, this will have a negative impact on the implementation of environmentally responsible activities in the whole supply chain and eventually create serious environmental problems. In terms of policy, by strengthening the environmental supervision of the core enterprises in the supply chain, governments can make use of the vertical spillover effect among supply chain enterprises to induce the enterprises in the supply chain to implement environmentally responsible activities, thereby improving the effectiveness of the government’s environmental governance.

This study has preliminarily confirmed the influence of supply chain power on environmental responsibility and the mediating effect of a company’s operating performance.

Of course, due to the uncontrollable nature of the data, this research has some limitations, such as the use of the net cash ratio to measure supply chain power and the sample of listed companies. Future research directions include: (1) survey methods could be used to obtain more supply chain data to verify the above findings; (2) social network methods could be used to construct supply chain networks to accurately measure the power of supply chain networks; and (3) studies could explore whether there is a non-linear influence between supply chain power and corporate environmental responsibility.

## Figures and Tables

**Table 1 ijerph-18-09264-t001:** Definition and calculation of the variables.

Variable Type	Variable Symbol	Calculation Method or Definition
Dependent variable	*CER*	2010–2018 Corporate environmental responsibility scores in Hexun.com’s Corporate Social Responsibility Report
Independent variable	*CR*	Actual net cash/(net profit + depreciation and amortization + financial expenses)
Control variables	*Size*	ln (Operating income)
	*Lev*	Total liabilities/total assets
	*Growth*	(Total net profit this year–total net profit last year)/total net profit last year
	*Dual*	This variable = 1 if the chairperson and general manager are the same person and 0 otherwise
	*Top1*	Shares of the largest shareholder/total number of shares
	*BM*	1/(P/B ratio)
	*SOE*	1 = a state-owned holding company; 0 = others
	*ListAge*	ln(Listing years + 1)

**Table 2 ijerph-18-09264-t002:** Descriptive statistics.

Variable	Mean	Std. Dev.	Min	Max	Observations
*CER*	5.063	5.067	−15	30	17,984
*CR*	0.321	12.439	−1107.647	623.956	18,000
*Size*	22.285	1.463	18.360	27.303	17,999
*Lev*	0.224	0.683	−0.686	7.781	17,672
*Growth*	0.217	0.412	0	1	17,713
*BM*	1.165	1.370	0.048	12.531	18,000
*Dual*	0.460	0.498	0	1	17,982
*SOE*	2.396	0.688	0	3.296	18,000
*ListAge*	0.343	0.152	0.083	0.755	18,000
*Top1*	0.063	0.140	−1.174	0.449	17,812
*ROA*	5.063	5.067	−15	30	17,984
*ROE*	−0.633	6.185	−365.087	27.204	18,000

**Table 3 ijerph-18-09264-t003:** Regression of supply chain power on environmental responsibility.

Variables	(1)	(2)
	*CER*	*CER*
*CR*	0.009 ***	0.055 ***
	(0.43)	(2.78)
*Size*	0.735 ***	0.682 ***
	(24.09)	(24.32)
*Lev*	−1.684 ***	−2.285 ***
	(−10.24)	(−15.21)
*Growth*	0.577 ***	0.481 ***
	(9.79)	(9.12)
*Dual*	−0.069	−0.010
	(−1.02)	(−0.17)
*BM*	0.288 ***	−0.112 ***
	(8.72)	(−3.65)
*SOE*	−0.471 ***	−0.063
	(−6.67)	(−0.98)
*ListAge*	0.513 ***	0.088 **
	(12.06)	(2.22)
*Top1*	0.983 ***	0.881 ***
	(4.82)	(4.74)
*Constant*	−11.265 ***	−11.069 ***
	(−17.82)	(−18.17)
*Year FE*	Yes	Yes
*Industry FE*	No	Yes
*Observations*	17,356	17,356
*R^2^*	0.093	0.278

Note: ** and *** are significant at 0.05 and 0.01, respectively.

**Table 4 ijerph-18-09264-t004:** Regression of supply chain power on the environmental responsibility of upstream suppliers.

Variables	(3)	(4)
	*CER*	*CER*
*CR^producer^*	0.089 ***	0.081 ***
	(3.00)	(2.79)
*Size*	0.063	0.414 ***
	(0.68)	(2.87)
*Lev*	−0.191	−0.073
	(−0.25)	(−0.09)
*Growth*	−0.495	−0.492
	(−0.96)	(−0.95)
*Dual*	0.176	0.283
	(0.58)	(0.91)
*BM*	−0.451	0.083
	(−0.64)	(0.11)
*SOE*	0.007	0.584
	(0.01)	(0.81)
*ListAge*	−0.096	−0.078
	(−0.48)	(−0.36)
*Top1*	−0.010 *	0.002
	(−1.71)	(0.23)
*Constant*	3.884 *	−2.477
	(1.89)	(−0.73)
*Year FE*	Yes	Yes
*Industry FE*	No	Yes
*Observations*	793	793
*R^2^*	0.036	0.184

Note: * and *** are significant at 0.1 and 0.01, respectively.

**Table 5 ijerph-18-09264-t005:** Regression of supply chain power on the environmental responsibility of upstream manufacturers.

Variables	(5)	(6)
	*CER*	*CER*
*CR^customer^*	0.033 **	0.032 **
	(2.42)	(2.44)
*Size*	−0.030	−0.187 **
	(−0.47)	(−2.17)
*Lev*	1.304 *	−0.359
	(1.81)	(−0.41)
*Growth*	0.264	0.326
	(0.52)	(0.66)
*Dual*	0.495	0.256
	(1.63)	(0.83)
*BM*	−1.040 *	−0.041
	(−1.74)	(−0.06)
*SOE*	0.714	0.881
	(0.99)	(1.24)
*ListAge*	−0.252	0.058
	(−1.54)	(0.34)
*Top1*	−0.002	0.006
	(−0.32)	(1.05)
*Constant*	5.104 ***	10.432 ***
	(4.01)	(5.32)
*Year FE*	Yes	Yes
*Industry FE*	No	Yes
*Observations*	1408	1408
*R^2^*	0.018	0.117

Note: *, ** and *** are significant at 0.1, 0.05 and 0.01, respectively.

**Table 6 ijerph-18-09264-t006:** Regression of supply chain power on business performance.

Variables	(7)	(8)
	*ROA*	*ROE*
*CR*	0.002 ***	0.003 ***
	(5.26)	(3.55)
*Size*	0.017 ***	0.039 ***
	(20.95)	(23.31)
*Lev*	−0.110 ***	−0.154 ***
	(−25.41)	(−16.10)
*Growth*	0.019 ***	0.043 ***
	(19.96)	(20.31)
*Dual*	0.001	0.001
	(0.65)	(0.46)
*BM*	−0.012 ***	−0.021 ***
	(−15.79)	(−11.82)
*SOE*	−0.006 ***	−0.012 ***
	(−3.79)	(−3.75)
*ListAge*	0.002	0.006 **
	(1.26)	(2.44)
*Top1*	0.026 ***	0.053 ***
	(5.30)	(5.52)
*Constant*	−0.289 ***	−0.727 ***
	(−16.93)	(−20.91)
*Year FE*	Yes	Yes
*Industry FE*	Yes	Yes
*Observations*	17,371	17,207
*R^2^*	0.291	0.209

Note: ** and *** are significant at 0.05 and 0.01, respectively.

**Table 7 ijerph-18-09264-t007:** Regression results of enterprise supply chain power, business performance, and environmental responsibility.

Variables	(9)	(10)
	*CER*	*CER*
*CR*	0.006	0.008
	(0.19)	(0.28)
*ROA*	16.729 ***	
	(20.52)	
*ROE*		9.267 ***
		(25.85)
*Size*	0.518 ***	0.415 ***
	(8.79)	(6.95)
*Lev*	−0.763 **	−0.834 ***
	(−2.43)	(−2.65)
*Growth*	0.104	0.009
	(1.42)	(0.13)
*Dual*	−0.071	−0.057
	(−0.59)	(−0.48)
*BM*	0.092	0.073
	(1.50)	(1.21)
*SOE*	−0.068	−0.071
	(−0.53)	(−0.55)
*ListAge*	0.394 ***	0.366 ***
	(3.69)	0.366 ***
*Top1*	0.282	0.188
	(0.73)	(0.49)
*Constant*	−9.181 ***	−6.712 ***
	(−7.54)	(−5.44)
*Year FE*	Yes	Yes
*Industry FE*	Yes	Yes
*Observations*	17,356	17,192
*R^2^*	0.322	0.355

Note: ** and *** are significant at 0.05 and 0.01, respectively.

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
