# Peer review of "Supply Chain Power and Corporate Environmental Responsibility: Mediation Effects Based on Business Performance"

_ijerph, 2021, doi:10.3390/ijerph18179264_

Round 1

Reviewer 1 Report

Dear authors, the article has improved substatiantilly, congratulations.

Minor: Eqs: 1, 2, 3 are not shown.

Author Response

Dear Reviewer 1:

    Thank you for yourletter and for the reviewers’ comments concerning ourmanu entitled “Supply Chain Power and Corporate Environmental Responsi-bility: Mediation Effects Based on Business Performance” (ID:ijerph-1288816).Those comments are all valuable and very helpful for revising andimproving our paper, as well as the important guiding significance toour researches. We have studied comments carefully and have madecorrection which we hope meet with approval. Revised portion aremarked in red in the paper. The main corrections in the paper and theresponds to the reviewer’s comments are as flowing:

Respondsto the reviewer’s comments:

Response tocomment: Minor: Eqs: 1, 2, 3 are not shown.

Response:

The equation in the article has been revised.

We have used MDPI's Editing service to polish the English language of our manuscript.

 Specialthanks to you for your good comments.

Reviewer 2 Report

This paper takes Chinese listed companies as a research sample and analyzes the impact of supply chain power on environmental responsibility. It is a good study, with a certain degree of innovation and enlightenment. For management practice, it has practical value. However, this paper still needs to be revised and improved from the following aspects:

  1. This paper studies the spillover effects of environmental responsibility in the supply chain, emphasizing the vertical spillover effects of upstream and downstream. However, corporate environmental responsibility also has horizontal spillover effects between neighboring companies. In this regard, some relevant documents need to be supplemented, such as:

Gu, J. Spatial Dynamics between Firm Sales and Environmental Responsibility: The Mediating Role of Corporate Innovation. Sustainability 2021, 13,1684. https://doi.org/10.3390/su13041684

For the horizontal spillover effects of corporate environmental responsibility, there should be a certain discussion, indicating that the author of this article has paid attention to and mastered relevant research, and pointed out that this article currently focuses on the research of vertical spillover effects. In this way, the positioning of the paper is very clear.

  1. Regarding the measurement of supply chain power, the paper introduces too little content. This is a key concept of this article. It is necessary to explain in detail why net cash ratio is an effective measure of supply chain power. (line 188)

  1. For the matching of upstream and downstream enterprises, this article has made some explanations, but it is not detailed (line 210-217). Since this part of the content is the core of this article, the result of the match will directly have a fundamental impact on the conclusion of the article. Therefore, it is necessary to introduce the matching details and results of upstream and downstream enterprises in detail to illustrate the scientificity and rationality of matching.

  1. The formula is missing (line 206,line 218 and line 209)

  1. The paper needs to provide a table of correlation coefficients(line 246-247).

  1. In Table 3, 4 and 5, why are there two models? Are they two different samples? It does not seem to be clearly stated in the paper. Therefore, it is necessary to explain the reasons for the juxtaposition of the two models in these tables.

  1. The format of quoting names should be uniform. For example, lines 91-95, jacobs & Singhal(2020) and osuji&nnodim (2008). The quotation of the person's name does not conform to the norm. This issue needs to be unified throughout the text.

jacobs & Singhal(2020)state that by considering the impact of 91 poor corporate social responsibility behaviour, this work needs to be extended from the 92 focal firm to partners and competitors up and down the supply chain[1].This is because 93 firms in the supply chain can influence other firms in the supply chain through power. 94 amaeshi, osuji&nnodim (2008) state that the more powerful firms in the supply chain can

8.Inthe note of Table 3, ** p<0.05, * p<0.1, can be deleted because they are not shown in Table 3 at all.

  1. This paper provides a certain explanation of the empirical research results, but the length is relatively short. It is recommended to add a discussion section to discuss the academic value and practical significance of this research.

Overall, this is a good paper. I believe that through the above improvements, the academic value of this paper will be even higher.

Author Response

Dear Reviewer 2:

    Thank you for yourletter and for the reviewers’ comments concerning ourmanu entitled “Supply Chain Power and Corporate Environmental Responsi-bility: Mediation Effects Based on Business Performance” (ID:ijerph-1288816).Those comments are all valuable and very helpful for revising andimproving our paper, as well as the important guiding significance toour researches. We have studied comments carefully and have madecorrection which we hope meet with approval. Revised portion aremarked in red in the paper. The main corrections in the paper and theresponds to the reviewer’s comments are as flowing:

Respondsto the reviewer’s comments:

  1. Response tocomment: This paper studies the spillover effects of environmental responsibility in the supply chain, emphasizing the vertical spillover effects of upstream and downstream. However, corporate environmental responsibility also has horizontal spillover effects between neighboring companies. In this regard, some relevant documents need to be supplemented, such as:

 Gu, J. Spatial Dynamics between Firm Sales and Environmental Responsibility: The Mediating Role of Corporate Innovation. Sustainability 2021, 13,1684. https://doi.org/10.3390/su13041684

 For the horizontal spillover effects of corporate environmental responsibility, there should be a certain discussion, indicating that the author of this article has paid attention to and mastered relevant research, and pointed out that this article currently focuses on the research of vertical spillover effects. In this way, the positioning of the paper is very clear.

Response:

In addition to the influence of the macro- and micro-factors on corporate environmental responsibility, scholars have also argued that the interrelationships between firms affect environmental performance. Gu (2021) argued that firm sales activities have a positive effect on environmental performance, with innovation having a mediating effect between the two. This is known as the environmental performance spatial spillover effect. The corporate environmental performance of a firm is influenced by its neighbours [8]. However, these studies focused on the horizontal spillover effect of corporate environmental responsibility. Therefore, the vertical spillover effect between upstream and downstream firms in the supply chain needs further study.

2.Response tocomment: Regarding the measurement of supply chain power, the paper introduces too little content. This is a key concept of this article. It is necessary to explain in detail why net cash ratio is an effective measure of supply chain power. (line 188)

 Response:

In addition to the influence of the macro- and micro-factors on corporate environmental responsibility, scholars have also argued that the interrelationships between firms affect environmental performance. Gu (2021) argued that firm sales activities have a positive effect on environmental performance, with innovation having a mediating effect between the two. This is known as the environmental performance spatial spillover effect. The corporate environmental performance of a firm is influenced by its neighbours [8]. However, these studies focused on the horizontal spillover effect of corporate environmental responsibility. Therefore, the vertical spillover effect between upstream and downstream firms in the supply chain needs further study.

3.Response tocomment: For the matching of upstream and downstream enterprises, this article has made some explanations, but it is not detailed (line 210-217). Since this part of the content is the core of this article, the result of the match will directly have a fundamental impact on the conclusion of the article. Therefore, it is necessary to introduce the matching details and results of upstream and downstream enterprises in detail to illustrate the scientificity and rationality of matching.

Response:

In order to study the vertical spillover effects in the supply chain, we constructed two matched samples, namely supply chain–producer and producer–customer samples. The CSMAR database provides data on the supply chain network relationship index. These data include supplier and customer data for listed companies. Firstly, producer and suppliers were screened. The supplier and producer codes were obtained and then the relevant variables’ values were obtained from CSMAR database. The environmental responsibility scores of the upstream suppliers were used as the dependent variable. Their related variables were used as control variables. The supply chain power of the downstream producer was used as the independent variable. Model 2 was then built, as shown below. Secondly, the manufacturers and customers were filtered. The manufacturer and customer codes were obtained and the values of the relevant variables were obtained from the CSMAR database. The upstream producer's environmental responsibility score was used as the dependent variable. The related variables were used as control variables and the supply chain power of the downstream customer was used as the independent variable (Model 3).

4.Response tocomment: The formula is missing (line 206,line 218 and line 209)

Response:

 The equation in the article has been revised.

5.Response tocomment: The paper needs to provide a table of correlation coefficients(line 246-247).

Response:

Appendix A. Variable Correlation Coefficient

CER

CR

Size

Lev

Growth

BM

Dual

SOE

ListAge

Top1

ROA

ROE

CER

1.000

CR

0.009

1.000

Size

0.259***

0.013**

1.000

Lev

0.104***

-0.009

0.463***

1.000

Growth

0.067***

0.004

0.023***

0.035***

1.000

BM

0.214***

0.007

0.682***

0.521***

-0.025***

1.000

Dual

-0.064***

-0.001

-0.184***

-0.158***

0.010

-0.138***

1.000

SOE

0.085***

0.009

0.331***

0.284***

-0.057***

0.262***

-0.288***

1.000

ListAge

0.098***

-0.005

0.323***

0.401***

0.001

0.245***

-0.239***

0.408***

1.000

Top1

0.070***

0.022***

0.191***

0.016***

0.005

0.078***

-0.038***

0.231***

-0.098***

1.000

ROA

0.174***

0.018***

-0.042***

-0.379***

0.175***

-0.206***

0.065***

-0.109***

-0.275***

0.127***

1.000

ROE

0.266***

0.019***

0.107***

-0.170***

0.190***

-0.054***

0.018***

-0.039***

-0.162***

0.131***

0.871***

1.000

6.Response tocomment: In Table 3, 4 and 5, why are there two models? Are they two different samples? It does not seem to be clearly stated in the paper. Therefore, it is necessary to explain the reasons for the juxtaposition of the two models in these tables.

Response:

Table 3 shows the regression results for Model 1, testing the impact of supply chain power on corporate environmental responsibility. A valid sample of 17,356 was obtained after removing missing values.

Table 4 shows the regression results for Model 2, testing for vertical spillover effects between upstream suppliers and downstream producers. After matching via the method described above, 793 valid samples were obtained.

Table 5 shows the regression results for Model 3, testing for vertical spillover effects between upstream producers and downstream customers. After matching via the method described above, 1408 valid samples were obtained.

7.Response tocomment: The format of quoting names should be uniform. For example, lines 91-95, jacobs & Singhal(2020) and osuji&nnodim (2008). The quotation of the person's name does not conform to the norm. This issue needs to be unified throughout the text.

 Response:

Jacobs and Singhal (2020) stated that this work needs to be extended from the focal firm to partners and competitors up and down the supply chain by considering the impact of poor corporate social responsibility behaviour [1]. This is because firms in the supply chain can influence other firms in the supply chain through power. Amaeshi, Osuji and Nnodim (2008) stated that the more powerful firms in the supply chain can exert some ethical pressure on the weaker party through codes of conduct, corporate culture, anti-pressure group campaigns, personnel training and value repositioning [9].

8.Response tocomment: Inthe note of Table 3, ** p<0.05, * p<0.1, can be deleted because they are not shown in Table 3 at all.

Response:

 Note: *, **, and *** are significant at 0.1, 0.05, and 0.01, respectively.

10.Response tocomment: This paper provides a certain explanation of the empirical research results, but the length is relatively short. It is recommended to add a discussion section to discuss the academic value and practical significance of this research.

Response:

In the past, research on corporate environmental responsibility has been conducted from within to outside the firm, arguing that the implementation of corporate environmental responsibility activities affects the firm's products and services, thus improving business performance. On the other hand, studies have argued that the pressures of factors external to the firm, such as laws, policies, market competition, consumers, etc., drive firms to implement environmentally responsible activities. In addition to these factors, the interactions between firms are also very important external factors that affect corporate environmental responsibility. The horizontal spillover effects between neighbouring firms have been studied. However, vertical spillovers from upstream and downstream enterprises have not received much attention. For example, companies may be required to have ISO 14000 environmental certification in order to become the suppliers of another firm. Therefore, the interaction between the upstream and downstream companies in the supply chain is also an important factor in the implementation of environmentally responsible activities by companies. This study constructed a panel data model for 2010–2018 using a sample of A-share listed companies in Shanghai and Shenzhen. The net cash ratio of the company was used as a proxy variable for supply chain power to discuss its impact on corporate environmental responsibility.

7.1. Theoretical Implications

Scholars have paid attention to the impact of external factors on corporate environmental responsibility. In fact, companies are no longer seen as independent individuals but are considered to interact with each other along the supply chain. Therefore, from a supply chain perspective, a firm's implementation of environmental responsibility activities must be influenced by other firms in the supply chain. This study analysed the impact of supply chain power on environmental responsibility. The main findings are as follows.

Firstly, supply chain power has a clear positive impact on corporate environmental responsibility. In a supply chain network, the higher a company's position in the supply chain network, the stronger its supply chain power and the more beneficial it is for companies to fulfil their environmental responsibility. Therefore, by occupying the core position in the supply chain network, enterprises can improve their business performance and help them fulfil their environmental responsibilities and achieve sustainable business operations.

Secondly, a firm's supply chain power can have a beneficial impact on the implementation of environmentally responsible activities by other companies in the supply chain, i.e., there is a vertical spillover effect. This finding suggests that supply chain power enhances the peer effect of supply chain firms.

Third, it was verified that business performance has a mediating effect between supply chain power and corporate environmental responsibility. The reason for the vertical spillover effect of supply chain power on corporate environmental responsibility is that supply chain power affects the business performance of enterprises. In order to improve their business performance, enterprises are bound to implement environmental responsibility activities in accordance with the requirements of the enterprises with supply chain power. In a supply chain, the firm with supply chain power is generally the core firm in the supply chain.

In conclusion, this study found that the supply chain power of a company is beneficial to the implementation of corporate environmental responsibility activities. Moreover, a firm's supply chain power has a vertical spillover effect on the implementation of environmental responsibility activities by other firms in the supply chain. Business performance plays an important mediating role between the two. These findings expand the field and scope of corporate environmental responsibility research. They enrich the theory of the external control of organisations from a supply chain perspective.

7.2. Managerial and Policy Implications

The findings have important managerial and policy implications. In terms of management, the core companies in a supply chain are generally the ones with supply chain power. The actions of core enterprises not only affect themselves but can also spill over vertically and affect other enterprises in the supply chain. If the core enterprises fail to effectively implement environmentally responsible activities, this will have a negative impact on the implementation of environmentally responsible activities in the whole supply chain and eventually create serious environmental problems. In terms of policy, by strengthening the environmental supervision of the core enterprises in the supply chain, governments can make use of the vertical spillover effect among supply chain enterprises to induce the enterprises in the supply chain to implement environmentally responsible activities, thereby improving the effectiveness of the government's environmental governance.

10.We have used MDPI's Editing service to polish the English language of our manuscript.

Specialthanks to you for your good comments.

Round 2

Reviewer 2 Report

This paper has reached the publication standard and can be published.

This manuscript is a resubmission of an earlier submission. The following is a list of the peer review reports and author responses from that submission.

Round 1

Reviewer 1 Report

I have following concerns for the authors to consider. Thanks.

1. The authors need to present the remaining sections of this paper at the end of the introduction section.

2. After reviewing the literature, the authors need to indicate the research gaps between the existing articles and the research topic. Then, the authors need to give their research works that bridge such gaps. Furthermore, the authors need to stress the contributions of their study.

3. The equations in this paper should be shown in a clearer way. Please define all the symbols (what are the parameters and what are the variables?) before the model.

4. The quantitative analysis and comparisons are simple. Are there any benchmarks that can be used to verify the effectiveness of the proposed method?

5. The authors need to summarize all their finding at the end of the computation.

6. At the conclusion section, the authors need to point out the limitations of their study and accordingly discuss future works in this regard.

Author Response

Dear Reviewers,

   Thanks a lot for your kind suggestion. Please see the modified one。The full text has been revised by native English speakers

  1. The authors need to present the remaining sections of this paper at the end of the introduction section.

The following content has been added:” The rest of this paper is arranged as follows: Part II: Literature Review; Part III: Theo-retical Analysis and Research Hypothesis; Part IV: Model and Data; Part V: Empirical Re-search Results and Analysis; Part VI: Robustness Analysis; Part VII: Test of the Mediating Effect of Business Performance; Part VIII: Conclusion.”

  1. After reviewing the literature, the authors need to indicate the research gaps between the existing articles and the research topic. Then, the authors need to give their research works that bridge such gaps. Furthermore, the authors need to stress the contributions of their study.

    The literature review section increases research gaps and contributions:”Corporate environmental responsibility is affected not only by individual corporate characteristics, macro-level factors and market competition factors but also by the rela-tionships between companies in the supply chain network. Although there have been studies on corporate social responsibility or environmental responsibility from a supply chain perspective, two issues must be further explored: (1) The direction and extent of the impact of supply chain network location on corporate environmental responsibility are not clear and ( 2) The effects of interaction between companies in different locations on the supply chain network are not clear. This article makes the following contributions: First, this work studies the impact on corporate environmental responsibility from the perspec-tive of supply chain network location, enriching the relevant research on the factors af-fecting corporate environmental responsibility. Second, this work clarifies the effect of po-sitioning in the supply chain network on business performance and environmental re-sponsibility and the intermediary effect of the two. Third, a method of empirical supply chain research is introduced, extending existing methods of supply chain research.”

  1. The equations in this paper should be shown in a clearer way. Please define all the symbols (what are the parameters and what are the variables?) before the model.

    The equation has been revised clearly. All variables and parameters are defined.”To test the impact of supply chain network location on corporate environmental re-sponsibility, corporate environmental responsibility is used as the explained variable and expressed as CER. Supply chain network location is the core explanatory variable, denot-ed as SCA. The main control variables include company size (Size), the debt-to-asset ratio (Lev), the operating income growth rate (Growth), two-time integration (Dual), the share-holding ratio of the largest shareholder (Top1), the book-to-market ratio (BM), the nature of property rights (SOE) and listing age (ListAge). We build the following model:

where i represents an individual enterprise, and t represents the year. Control stands for controlled variables, Industry stands for industry fixed effects, and Year stands for time fixed effects. The specific meanings of the variables and data sources are given in the next section. The parameters α0, α1 and α2 represent the intercept, core explanatory variable co-efficient and control variable coefficient, respectively.”” In Equation 2,the parameters λ0, λ1 and λ2 represent the intercept, core explanatory varia-ble coefficient and control variable coefficient, respectively. In Equation 3, the parameters γ0, γ1, γ2 and γ3 represent the intercept, core explanatory variable coefficients, intermediate variable coefficients, and control variable coefficients, respectively.”

  1. The quantitative analysis and comparisons are simple. Are there any benchmarks that can be used to verify the effectiveness of the proposed method?

    The Hausman test was used to test the model. “The Hausman test was used to determine whether to use a fixed effects model or a ran-dom effects model. The P value of the modified hausmanHausman test result is less than 0.000 (chi2(9)=483.57 depends on your choicethe selection of a P value accuracy level), in-dicating that a fixed effects model is required for empirical analysis.”

  1. The authors need to summarize all their finding at the end of the computation.

    We summarize all our finding at the end.” First, the location ofpositioning the supply chain network has a significant positive impact on corporate environmental responsibility. In the supply chain network, the higher the a firm’s position of the company in the supply chain network is, the stronger the its competitiveness of the company, and the better easier it is for the company to fulfill its en-vironmental responsibilities. Therefore, by occupying the central position of the supply chain network, enterprises can improve their operating performance and help them fulfill their environmental responsibilities and realize their sustainable operations.

Second, companies with a higher positioned higher in the supply chain network can have a positive impact on the environmental responsibilities of upstream and down-stream companies through the supply chain network. Enterprises with higher positions in the supply chain network can use their influence in the supply chain to influence affect the environmental responsibilities of upstream and downstream enterprises through the supply chain network. This finding verifies that the contagion effect between companies in the supply chain will not only have has a financial impact, but will also have an im-pact on the behavior relations between companies.

Third, it our results verifies verify the mediating effect of business perfor-mance between supply chain location and corporate environmental responsibility. Using ROA and ROE as the variables of business performance, a stepwise regression method and bootstrap test was were used to verify the mediating effect of business performance between the two, and the Bootstrap method was used to further verify the mediating effect between the two. The results show that the higher a company's position in the supply chain is, the better the company's operating performance becomes, and the better easier it is becomes for the company to fulfill its environmental responsibilities.”

  1. At the conclusion section, the authors need to point out the limitations of their study and accordingly discuss future works in this regard.

we  point out the limitations of  study and accordingly discuss future works in three  regards..” Of course, due to the inaccessibility of data, we use supply chain capital and the ac-quisition rate as measures of a company's position in the supply chain network and use a sample of listed companies, which may have certain limitations. In terms of future re-search directions, (1) investigations can obtain more supply chain data to verify our con-clusions; (2) when data become available, social networks can be used to construct a sup-ply chain network to accurately measure supply chain network positioning and (3) stud-ies can determine whether there is a nonlinear effect between supply chain network posi-tioning and corporate environmental responsibility.”

Thanks again a lot for your kind suggestion.

Reviewer 2 Report

Dear Authors,

In my opinion you are sending a first draft of your work, which you have not read or corrected.
There are many repetitions throughout the document.
On the other hand, I recommend that when you make a general statement you base it on previous studies.
I also recommend that you review the title since there is no mediating effect model.
In the introduction you do not indicate the research question.
Please review and clarify the variables, it would be advisable to use a table.
Regarding the correlation, apart from the fact that the paragraph is repeated, this indicator does not measure multicollinearity, for which you would have to use methods such as the VIF method.
The results of so many regressions are not clear and I see that there are models that are different but have the same coefficients, for example 2 and 3.
Regarding the data the sample size varies which is not explained within the model which requires the same sample size.
Report that they have eliminated data due to outliers, without explaining what they consider outliers or in which variables they occur.
Minor failures, they indicate that they have not received funds and in the function of one of the authors it is indicated that he was in charge of obtaining funds.

Author Response

Dear Reviewers,

    Thanks a lot for your kind suggestion. Please see the modified one。The full text has been revised by native English speakers

1.In my opinion you are sending a first draft of your work, which you have not read or corrected.

There are many repetitions throughout the document.

Thanks a lot for your kind suggestion. The full text has been read and revised. The duplicates have been deleted. The full text has been revised by a native English speaker

On the other hand, I recommend that when you make a general statement you base it on previous studies.

The literature review section increases research gaps and contributions:”Corporate environmental responsibility is affected not only by individual corporate characteristics, macro-level factors and market competition factors but also by the rela-tionships between companies in the supply chain network. Although there have been studies on corporate social responsibility or environmental responsibility from a supply chain perspective, two issues must be further explored: (1) The direction and extent of the impact of supply chain network location on corporate environmental responsibility are not clear and ( 2) The effects of interaction between companies in different locations on the supply chain network are not clear. This article makes the following contributions: First, this work studies the impact on corporate environmental responsibility from the perspec-tive of supply chain network location, enriching the relevant research on the factors af-fecting corporate environmental responsibility. Second, this work clarifies the effect of po-sitioning in the supply chain network on business performance and environmental re-sponsibility and the intermediary effect of the two. Third, a method of empirical supply chain research is introduced, extending existing methods of supply chain research.”

2.I also recommend that you review the title since there is no mediating effect model.

    We have revised the title of the article.” Supply Network Location and Corporate Environmental Re-sponsibility: Mediation Effect Based on Business Performance”

3.In the introduction you do not indicate the research question.

    Added research questions in the introduction .“To compensate for the deficiencies of the above approach, this article mainly studies the following three issues: (1) the impact of supply chain network location on corporate envi-ronmental responsibility; (2) the impact of supply chain network location on the environ-mental responsibility of other companies in the supply chain network; and(3) whether business performance is between the supply chain network location and environmental responsibility plays a mediating role.”

Please review and clarify the variables, it would be advisable to use a table.

All variables have been clarified using the table.

Table 1 Variable definition and calculation

Variable type

Variable symbol

Variable calculation method or definition

explained variable

CER

2010-2018 Corporate Environmental Responsibility Score provided in Hexun.com Corporate Social Responsibility Reports

explanatory variable

SCA

(Accounts payable + accounts received in advance-accounts receivable-accounts paid in advance-inventory)/operating income

CR

Cash flow generated in operating activities/(net profit + depreciation and amortization + financial expenses)

Size

ln(Operating income)

control variable

Lev

Total liabilities/total assets

Growth

(Total net profit this year-total net profit last year)/Total net profit last year

Dual

If the chairman and general manager are the same person, it is 1, otherwise it is 0

Top1

Largest shareholder/total number of shares

BM

1/(P/B ratio)

SOE

The value of a state-owned holding company is 1. Others fiems are assigned a value of 0

ListAge

ln(Listing years+1)

4.Regarding the correlation, apart from the fact that the paragraph is repeated, this indicator does not measure multicollinearity, for which you would have to use methods such as the VIF method.

The results of so many regressions are not clear and I see that there are models that are different but have the same coefficients, for example 2 and 3.

    (1) After the indicators were tested for correlation, the VIF method was used to test for multicollinearity.” To test for the possible multicollinearity problems in the model, we has carried out a correlation analysis of the variables. The results show that the correlation coefficients be-tween the explanatory and control variables of the model are less than 0.5. The largest test VIF value is 2.25..These reslults showthat no serious multicollinearity problems affect the regression model explored in this paper.”

    (2) The regression results in the article have been reconfirmed and the relevant results have been revised. Please see the modified one。

5.Regarding the data the sample size varies which is not explained within the model which requires the same sample size.

Because it is necessary to match the upstream and downstream companies with listed companies and delete the data of non-listed companies, there are differences in sample data.”The above research only studies the impact of supply chain capital on corporate en-vironmental responsibility from the perspective of the enterprise itself, and does not reflect the relationship between the location of upstream and downstream enterprises in the supply chain and environmental responsibility. Therefore, the supply chain network rela-tional database of listed companies provided by Guotaian Data is used to construct a first-level supply chain, and the influence of the network location of downstream enter-prises in the supply chain on the environmental responsibility of upstream enterprises is studied. The manufacturers included in this database are listed companies. As data on nonlisted companies are difficult to obtain, it is necessary to select suppliers or customers from the database that are also listed companies to match the first-level supply chain, and eliminate the data of suppliers and nonlisted companies. Supply chain network position-ing is measured by the ratio of upstream companies to downstream companies using formulas (â… ) and (â…ˇ):

SCA I represents the ratio of supplier and manufacturer supply chain capital, 〖SCA〗_supplyrepresents the supplier's supply chain capital, and〖SCA〗_producer represents the producer's supply chain capital. CRI represents the ratio of the supplier's cash-receiving rate to the producer, 〖CR〗_supply represents the supplier's cash-receiving rate, and 〖CR〗_producerrepresents the producer's cash-receiving rate. SCA II represents the ratio of the manufacturer’s and customer’s supply chain capital, 〖SCA〗_producerrepresents the producer’s supply chain capital, and 〖SCA〗_customerrepresents the customer’s supply chain capital. CRâ…ˇ represents the ratio of the producer’s to the customer's cash-receiving rate, 〖CR〗_producer represents the producer's cash-receiving rate, and〖CR〗_customer  represents the customer's cash-receiving rate. After excluding data on nonlisted companies, the sample size of SCA I is 729, the sample size of CR I is 793, the sample size of SCA II is 1335, and the sample size of CR II is 1408.

Report that they have eliminated data due to outliers, without explaining what they consider outliers or in which variables they occur.”

6.Minor failures, they indicate that they have not received funds and in the function of one of the authors it is indicated that he was in charge of obtaining funds.

    Funding: Please add: This research was funded by the National Social Science Foundation of China (grant number 17BJY102)

Thanks again a lot for your kind suggestion.

Round 2

Reviewer 1 Report

The authors revised the manuscript according to the first round comments. I appreciate a lot for the authors’ efforts, but still have some concerns as follows:

  1. The authors need to clarify the method they used to incorporate the environmental responsibility into the supply chain/logistics/transportation management. Typically, multi-objective optimization (e.g., Wang et al., https://doi.org/10.1016/j.dss.2010.11.020), emission cap-and-trade (e.g., Fang et al., https://doi.org/10.3390/su9122198), and emission tax (e.g., Sun et al., https://doi.org/10.1155/2018/8645793) are the commonly used methods in dealing with the environmental issue. Therefore, referring to above literature, the authors need to present their consideration.
  2. It is very strange to see “supply chain location”. Usually, “location” is associated with nodes (e.g., factories or distribution centers) in the supply chain. Therefore, the authors need to explain it or use other definitions to represent what they want to express.
  3. The hypotheses in this paper are drawn from existing literature based on the authors’ description. So, is it meaningful to examine them again in this paper?
  4. The model in Section 4.1 is still not well explained. Please define “controli,t”, “Σindustry”, “Σyear”, and “εi,t
  5. What is “supply chain ownership”?
  6. Please give the motivation why robustness test is needed.

Other concerns:

  1. The format of this paper should be edited according to the journal’s style.
  2. Please check the language of this paper carefully.

Reviewer 2 Report

Dear authors, I still see many nonsenses in your article, for example:

 line 524 Therefore, by occupying the central position of the supply chain network, enterprises can improve their operating performance fulfill their environmental responsibilities and realize their sustainable operations.